# Gram-scale total synthesis of teixobactin promoting binding mode study and discovery of more potent antibiotics

Yu Zong[1,5], Fang Fang[1,5], Kirsten J. Meyer[2], Liguo Wang[1], Zhihao Ni[1], Hongying Gao[3], Kim Lewis[2], Jingren Zhang[4] & Yu Rao[1]

Teixobactin represents a new class of antibiotics with novel structure and excellent activity against Gram-positive pathogens and *Mycobacterium tuberculosis*. Herein, we report a one-pot reaction to conveniently construct the key building block L-allo-Enduracidine in 30-gram scale in just one hour and a convergent strategy $(3 + 2 + 6)$ to accomplish a gram-scale total synthesis of teixobactin. Several analogs are described, with **20** and **26** identified as the most efficacious analogs with 3~8-fold and 2~4-fold greater potency against vancomycin resistant *Enterococcus faecalis* and methicillin-resistant *Staphylococcus aureus* respectively in comparison with teixobactin. In addition, they show high efficiency in *Streptococcus pneumoniae* septicemia mouse model and neutropenic mouse thigh infection model using methicillin-resistant *Staphylococcus aureus*. We also propose that the antiparallel β-sheet of teixobactin is important for its bioactivity and an antiparallel dimer of teixobactin is the minimal binding unit for lipid II via key amino acids variations and molecular docking.

[1] MOE Key Laboratory of Protein Sciences, School of Pharmaceutical Sciences, MOE Key Laboratory of Bioorganic Phosphorus Chemistry & Chemical Biology, Tsinghua University, 100084 Beijing, China. [2] Antimicrobial Discovery Center, Northeastern University, Department of Biology, Boston, MA 02115, USA. [3] Tsinghua-Peking Center for Life Sciences, Haidian District, 100084 Beijing, China. [4] Center for Infectious Disease Research, School of Medicine, Tsinghua University, Beijing, China. [5]These authors contributed equally: Yu Zong, Fang Fang. Correspondence and requests for materials should be addressed to Y.R. (email: yrao@tsinghua.edu.cn)

Antibiotic resistance becomes a rapidly growing health concern and poses great challenge for the human[1–3]. In the postantibiotic era, people may die from ordinary infections and minor injuries due to a lack of effective antibiotics, so novel antibacterial lead structures are urgently needed for guaranteeing future therapeutic efficacy. Recently, a new antibiotic named teixobactin was reported by Lewis and coworkers, and it exhibits excellent bioactivities against Gram-positive pathogens and *Mycobacterium tuberculosis*[4]. Given that teixobactin targets conserved substrates (Lipid II and III) rather than enzymes in peptidoglycan and teichoic acid biosynthetic pathways, the emergence of resistance is expected to be difficult, which makes teixobactin a promising candidate for further drug development[5].

Teixobactin contains a nonnatural amino acid L-*allo*-End possessing a unique cyclic guanidine moiety, which exists in other natural products, such as mannopeptimycin and enduracidins[6,7]. However, the presence of the unnatural amino acid L-*allo*-End complicates its total synthesis as reflected in several endeavors from the Payne[8], Li[9], and Chen[10] groups. They achieved the milligram scale total synthesis of teixobactin. Payne reported the first total synthesis of teixobactin in global solid-phase peptide synthesis (SPPS). Simultaneously, Li presented an elegant total synthesis of teixobactin using Ser/Thr ligation. Chen achieved a synthesis with a linear synthetic procedure recently. Despite these impressive developments, scalable synthesis of teixobactin has remained elusive, because of the tedious synthesis of L-*allo*-End and generation of diketopiperazine (DKP) by-product in SPPS[8] (Fig. 1). On the other hand, the L-*allo*-End seems to be important for biological activity. When L-*allo*-End was replaced by Arg or Lys, it led to about fourfold loss of potency[11–15]. So far, by replacement of L-*allo*-End and modifications on other amino acids, only Singh and our group reported analogs with equal potency to teixobactin[16–18]. However, it was a pity that any bioactivity of analogs retaining L-*allo*-End has not been reported. Therefore, further probing the structure-activity relationship of teixobactin analogs retaining L-*allo*-End to gain stronger antibiotics is valuable. Herein, we demonstrate a one-pot reaction to conveniently construct the key L-*allo*-End building block in 30-g

scale and a convergent strategy to achieve gram-scale synthesis of teixobactin. A series of analogs retaining L-*allo*-End are synthesized and more potent candidates are discovered. Finally, a detailed binding model in which the antiparallel dimer of teixobactin is the minimal binding unit for lipid II and the antiparallel β-sheet of teixobactin is important for its bioactivity is raised via key amino acids variations and docking.

## Results and discussion

**Retrosynthetic analysis of teixobactin.** Initially, we attempted to accomplish the total synthesis of teixobactin in a convergent synthetic route in which we could readily change any amino acids to access teixobactin analogs without resorting to de novo synthesis in global SPPS. Simultaneously, drawing lessons from our previous procedure of yielding a series of teixobactin analogs by combining solution-phase with solid-phase synthesis, we proposed that teixobactin could also be derived from a cyclic pentapeptide and a linear hexapeptide which could be constructed in solution-phase and solid-phase, respectively ("5 + 6")[18]. During the removal of Fmoc of L-*allo*-End residue in SPPS, the generated free α-amine could attack the adjacent α-carboxyl of L-isoleucine to form unwanted DKP readily[8]. To avert this by-product, dipeptide of Alloc-Ala-*allo*-End-OH was considered as an integral unit in the synthesis of cyclic pentapeptide. In addition, as ester condensation between Ile and secondary OH group of D-Thr in SPPS led to 33% epimerization[19], we intended to achieve the esterification in solution phase to exclude this side product. So the cyclic pentapeptide could be divided into dipeptide and tripeptide ("2 + 3") (Fig. 1).

**Synthesis of L-*allo*-End building block.** Presently, there are three original routes to approach L-*allo*-End. Du Bois group used rhodium catalyst to provide racemic End from compound **1**[20]. Payne group constructed L-*allo*-End in seven steps from Boc-Asp-OtBu (compound **2**)[8]. The Yuan group built L-*allo*-End in ten steps from hydroxyproline (compound **3**) in 31% overall yield[21] (Fig. 2). We sought a more concise synthetic strategy for

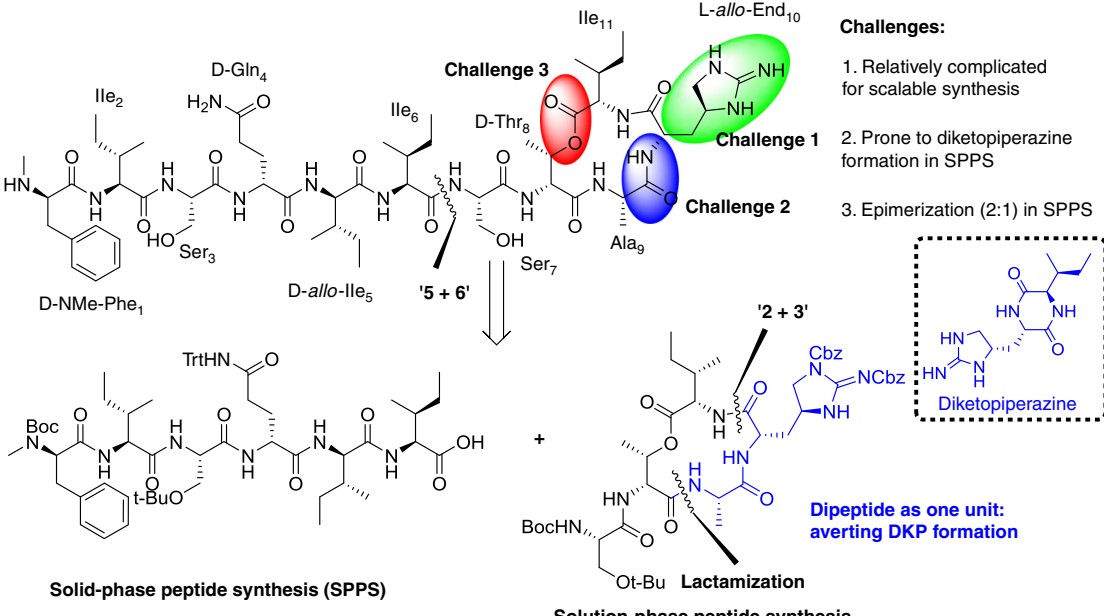

**Fig. 1** Current challenges and retrosynthetic analysis of teixobactin. There are three challenges in scalable total synthesis of teixobactin. Teixobactin could be synthesized via the convergent strategy (3 + 2 + 6)

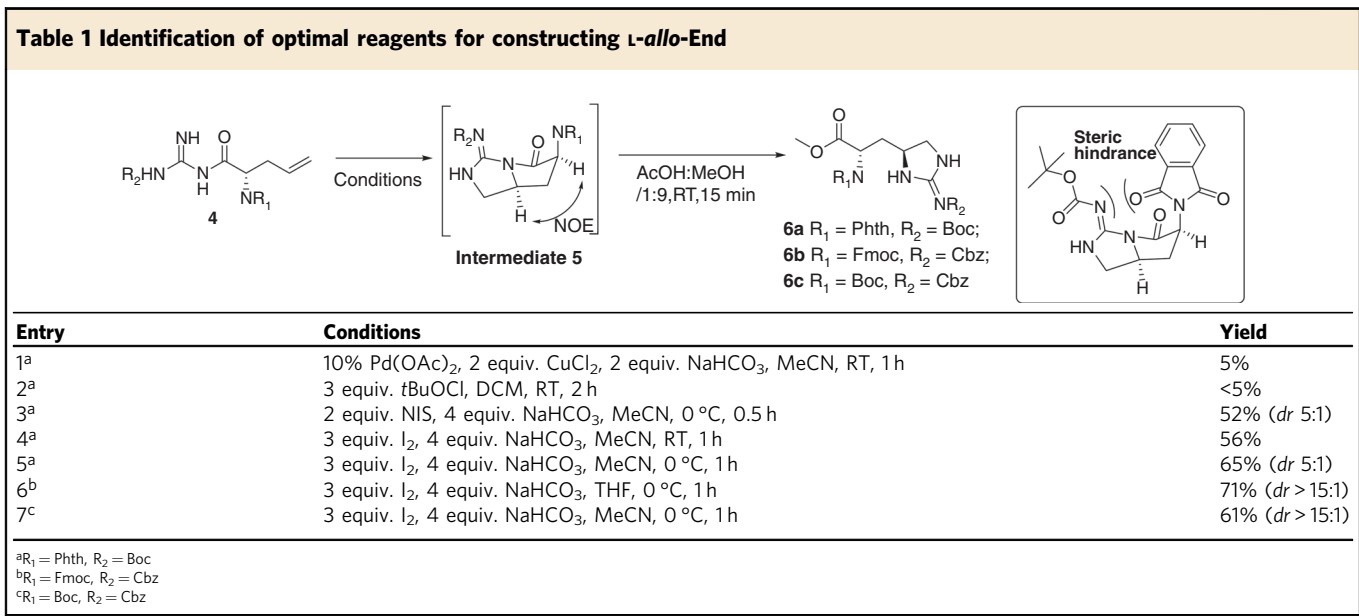

**Fig. 2** Synthetic strategy to L-*allo*-enduracididine. There are three reported routes to provide End. Our synthetic route just takes 1 h via a one-pot reaction to synthesize desired End

---

**Table 1 Identification of optimal reagents for constructing L-*allo*-End**

| Entry | Conditions | Yield |
|---|---|---|
| 1[a] | 10% Pd(OAc)$_2$, 2 equiv. CuCl$_2$, 2 equiv. NaHCO$_3$, MeCN, RT, 1 h | 5% |
| 2[a] | 3 equiv. *t*BuOCl, DCM, RT, 2 h | <5% |
| 3[a] | 2 equiv. NIS, 4 equiv. NaHCO$_3$, MeCN, 0 °C, 0.5 h | 52% (*dr* 5:1) |
| 4[a] | 3 equiv. I$_2$, 4 equiv. NaHCO$_3$, MeCN, RT, 1 h | 56% |
| 5[a] | 3 equiv. I$_2$, 4 equiv. NaHCO$_3$, MeCN, 0 °C, 1 h | 65% (*dr* 5:1) |
| 6[b] | 3 equiv. I$_2$, 4 equiv. NaHCO$_3$, THF, 0 °C, 1 h | 71% (*dr* > 15:1) |
| 7[c] | 3 equiv. I$_2$, 4 equiv. NaHCO$_3$, MeCN, 0 °C, 1 h | 61% (*dr* > 15:1) |

[a]R$_1$ = Phth, R$_2$ = Boc
[b]R$_1$ = Fmoc, R$_2$ = Cbz
[c]R$_1$ = Boc, R$_2$ = Cbz

---

providing adequate L-*allo*-End to simplify the total synthesis of teixobactin and promote potential clinical applications.

Inspired by inter- and intramolecular diamination and guanidinylation[22–26] of alkenes, we proposed a cascade reaction in which intramolecular guanidinylation of compound **4** occurred to generate intermediate **5**, followed by alcoholysis of the amide bond to access protected L-*allo*-End (Fig. 2). Our design required preinstalled guanidine at C-terminal via Bop-mediated amide condensation, followed by cycloguanidylation and alcoholysis. After a preliminary screening, palladium (II) or *t*-BuOCl was infeasible to efficiently afford the desired product (entries 1 and 2). In contrast, NIS and I$_2$ could generate desired product in 52% and 56% yields, respectively (entries 3 and 4). When the temperature was reduced to 0 °C, I$_2$ mediated cycloguanidylation furnished the desired product in 65% isolated yield but with moderate stereoselectivity (*dr* = 5:1) which was presumably due to the weak steric hindrance between R$_1$ (Boc) and R$_2$ (Phth) (entry 5). When the protecting groups were altered (R$_1$ = Fmoc, R$_2$ = Cbz or R$_1$ = Boc, R$_2$ = Cbz), products **6b** and **6c** could be

separated in 71% and 61% yield, respectively with excellent stereoselectivity (*dr* > 15:1) (entries 6 and 7). Arising from the instability of intermediate **5**, alcoholysis occurred readily to furnish desired compound **6**. It just took 1 h for this one-pot reaction to access protected L-*allo*-End (Table 1)

As intermediate **5a** was not separable for its instability, the relatively stable compound **7** without Boc was separated and confirmed through nuclear magnetic resonance (NMR) analysis. Proposed structure of intermediate **5a-2** was indirectly confirmed by NOE between α and γ position. Meanwhile, we detected the existence of three kinds of intermediates through liquid chromatography–mass spectrometry and NMR data analysis (Fig. 3). The $^1$H NMR data of free L-*allo*-End (compound **8**) derived from compound **6b** was in good agreement with literature reports[20,21]. The building block **6b** could be obtained in 30-g scale and it was favorable for scalable total synthesis of teixobactin and SAR study. The guanidine group of compound **6b** was further protected with Cbz to obtain compound **9** to exclude side reaction on guanidine in the following reactions (Fig. 4).

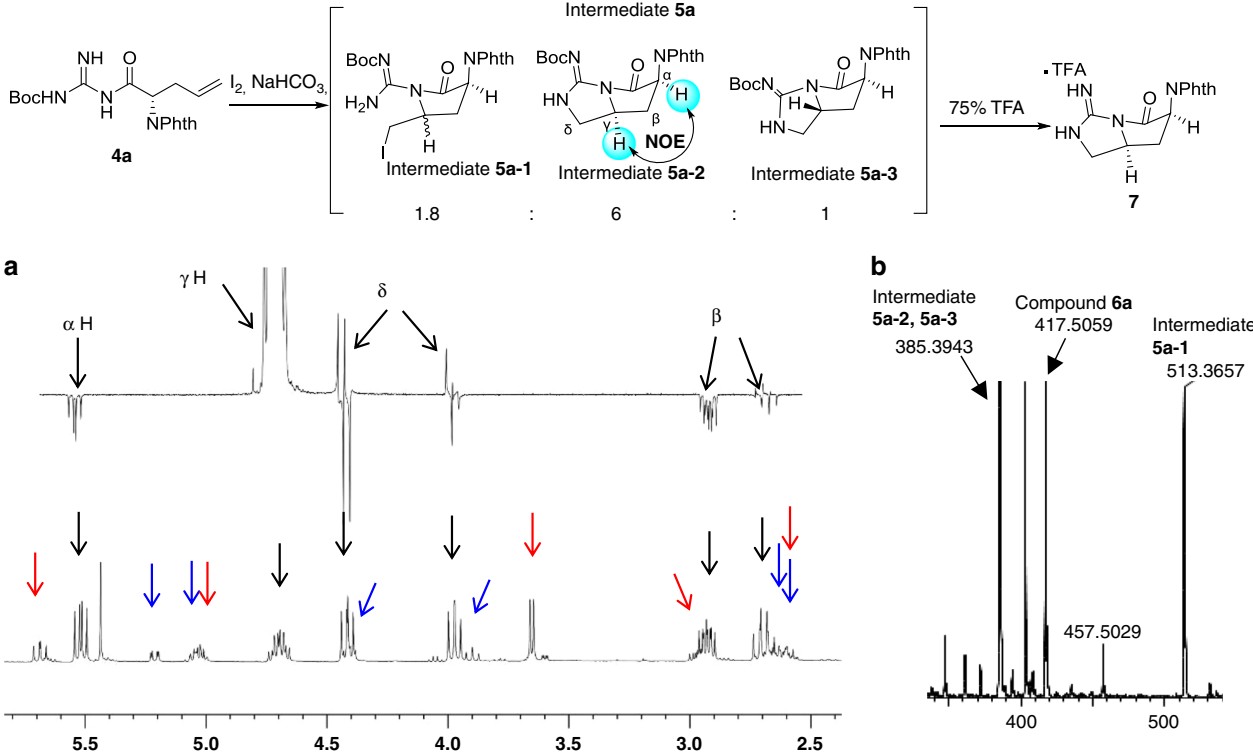

**Fig. 3** $^1$H NMR spectrum of intermediates **5**. **a** NOE and $^1$H NMR for intermediates **5a**. The red arrow indicates the peaks of intermediate **5a-1**. The black arrow indicates the peaks of intermediate **5a-2**. The blue arrow indicates the peaks of intermediate **5a-3**. **b** MS data for intermediates **5a** (LC–MS analysis using MeOH as a solvent)

**Total synthesis of teixobactin**. With adequate L-*allo*-End in hand, we embarked on gram scale total synthesis of teixobactin with the proposed convergent strategy (3 + 2 + 6). Starting from protected D-Threonine **10**, Boc was removed in 30% trifluoroacetic acid, followed by condensation with Boc-Ser(*t*Bu)-OH to furnish dipeptide **11** without purification. Then the coupling of compound **11** with Fmoc-Ile-OH proceeded, followed by removal of Fmoc to provide tripeptide **12** in 79% yield. As mentioned, undesired DKP[27] can appear during removal of Fmoc of L-*allo*-End in SPPS. So the dipeptide Alloc-Ala-*allo*-End(Cbz)$_2$-OH could be used as an integral building block to circumvent the side product. After removal of Fmoc from compound **9**, the condensation was completed with Alloc-Ala-OH to afford dipeptide **13**. Then the ester bond of compound **13** was hydrolyzed by LiOH at room temperature to obtain compound **14** in mild yield. It was noticed that during this process, 25% of Cbz was cleaved. When the temperature was reduced to 0 °C, this hydrolysis was completed in 4 min and we nearly did not observe significant appearance of side product **14-1**. Then we attempted to use HATU to promote the condensation of dipeptide **14** and tripeptide **12**. However, side product DKP **15-1** was detected during the process. It is worthwhile noting that the coupling reagent DEPBT[28] could completely avert the occurrence of intramolecular cyclization to generate linear pentapeptide **15** in 63% yield. After the allyl and alloc of intermediate **15** were removed in the presence of Pd(PPh₃)₄/1,3-dimethylbarbituric acid, final cyclization occurred with HATU/HOAt/DIEA[29] to afford compound **16** in 58% yield. Linear hexapeptide (compound **17**) was readily prepared in SPPS[18]. Then the cyclic pentapeptide and linear hexapeptide were coupled together with DEPBT/DIEA[30] to generate protected teixobactin **18** without purification. Cbz was removed under reducing conditions (Pd(OH)₂/C, H₂) and final global deprotection was subsequently carried out with TFA–H₂O–TIPS (95:2.5:2.5) in 1 h. The crude peptide was purified by semi-preparative reversed-phase high-performance liquid chromatography (HPLC) to furnish teixobactin **19** in 31% yield from compound **16** and in 8% overall yield from compound **9** (Fig. 4).

**In vitro efficacy**. Previously, Su group[12] and our group[18] have demonstrated that the methyl group on *N*-Me-D-Phe₁ was not essential, so a series of analogs reserving L-*allo*-End and lacking the methyl group were synthesized via the our synthetic route. Recently, Tajkhorshid group reported that residue *N*-Me-D-Phe₁ came into contact with the membrane surface via molecular dynamics simulation[31]. In our previous study based on Lys₁₀-teixobactin, we observed that increased hydrophobicity of Phe₁ could improve bioactivity[18]. To increase the ability of anchoring on the membrane by teixobactin, we constructed four analogs **20**–**23** with different modifications at the N-terminus. Compared with teixobactin (minimum inhibitory concentration (MIC) value is 0.5 μg ml⁻¹), compounds **21**–**23** gave MIC value of 0.125 μg ml⁻¹, and compound **20** gave lower MIC value of 0.09 μg ml⁻¹ against VRE TH4938 (Fig. 5). Specifically, as benzophenone was a good photoaffinity group for covalent labeling of protein–probe complexes[32,33], compound **21** was a potential probe for target identification. In order to balance the hydrophobicity and hydrophilicity of compound **20**, D-Gln₄ was varied to D-Arg₄ to access compound **24**, but it led to twofold loss of activity (MIC value is 0.25 μg ml⁻¹) (Fig. 5). According to the molecular dynamics simulations[31,34] and solid NMR study[35], amide groups on the cyclodepsipeptide (D-Thr₈-Ile₁₁) ring backbone formed H-bond with negatively charged pyrophosphate group of lipid II (Please see below). In addition, based on our previous study of modification of Lys₁₀-teixobactin, we found that bioactivity of lactam was slightly better than lactone against MRSA[18]. To further increase hydrogen bonding interaction

**Fig. 4** Gram-scale total synthesis of teixobactin. **a** (1) 4 equiv. I$_2$, 5 equiv. NaHCO$_3$, 0 °C, 1 h; (2) MeOH/AcOH (9:1), 30 °C, 15 min, 66% for 2 steps; **b** (1) 3 equiv. LiOH, THF/H$_2$O; (2) Pd(OH)$_2$/C, H$_2$, MeOH/AcOH (9:1); **c** 3 equiv. CbzOSu, 4 equiv. DIEA, DCM, 30 °C, 4 h, 80%; **d** (1) 30 % TFA, 30 °C, 30 min; (2) 1.2 equiv. Boc-Ser(tBu)-OH, 1.2 equiv. HATU, 3 equiv. DIEA, DCM/DMF, 30 °C, 3 h; **e** (1) 1.5 equiv. Fmoc-Ile-OH, 1.5 equiv. EDCI, 0.2 equiv. DMAP, DCM, 12 h; (2) 33% Et$_2$NH, 30 °C, 15 min; 79%; **f** (1) 33% Et$_2$NH, r.t., 10 min; (2) 3 equiv. Alloc-Ala-OH, 3 equiv. HATU, 3 equiv. DIEA, DCM/DMF, 30 °C, 3 h, 71%; **g** 2 equiv. LiOH, THF/H$_2$O (3:1), 0 °C, 4 min; **h** 2.5 equiv. compound **12**, 2 equiv. DEPBT, 2.5 equiv. DIEA, THF/DMF, 30 °C, overnight, 63%; **i** (1) 0.3 equiv. Pd(PPh$_3$)$_4$, 2 equiv. 1,3-dimethylbarbituric acid, DCM, 30 °C, 1 h; (2) 4 equiv. HATU, 4 equiv. HOAT, 8 equiv. DIEA, DCM/DMF, 30 °C, 24 h, 58%; **j** (1) 3 M HCl, 15 min; (2) 1 equiv. compound **17**, 1.2 equiv. DEPBT, 1.2 equiv. DIEA, DMF,12 h; **k** (1) Pd(OH)$_2$/C, H$_2$, MeOH/HCOOH, 1 h; (2) TFA:TIPS: H$_2$O: 95:2.5:2.5. 31% from compound **16**

between the cyclic tetrapeptide (residues 8–11) and the pyrophosphate of lipid II, we intended to acquire analog **25** in which the lactone skeleton was replaced by lactam. According to the same procedures (3 + 2 + 6), the desired analog **25** was synthesized and exhibited slightly higher potency than teixobactin. Then compound **26** featured with N-terminal biphenyl group as well as C-terminal lactam scaffold was obtained and had a MIC of 0.0625 µg ml$^{-1}$, eight times better than teixobactin (Fig. 5, Supplementary Table 1). Moreover, these analogs were evaluated against different kinds of Gram-positive pathogens, particularly for drug-resistant strains. In comparison with teixobactin, compounds **20** and **26** were identified as the most potent analogs which exhibited up to sixfold and threefold greater potency against vancomycin-resistant enterococci (VRE TH4937) and methicillin-resistant *Staphylococcus aureus* (MRSA BAA-1695), respectively (Table 2). In addition, compound **20** gave minimum bactericidal concentration (MBC) value of 0.125 µg ml$^{-1}$, four times lower than teixobactin against MRSA BAA-1695 (Supplementary Table 2). In the time-dependent killing assay against *Enterococcus faecalis* ATCC29212, compound **20** was more potent than teixobactin. The effect of compound **20** (2.5 µg ml$^{-1}$) was as good as teixobactin (10 µg ml$^{-1}$) (Fig. 6a).

**In vivo efficacy**. The in vivo evaluation of compounds **20** and **26** was performed with mouse models[4]. Analogs **20** and **26** showed nearly no mammalian cytotoxicity against HepG2 cell line (Supplementary Fig. 2). B6 female mice were intraperitoneally infected by *Streptococcus pneumonia* at a dose that leads to 90% mortality. An hour post infection, compounds **20** and **26** were introduced i.v. at a single dose (0.5 mg kg$^{-1}$), respectively. Vancomycin was introduced i.v. at a dose of 5 mg kg$^{-1}$ as positive control. Treatment groups had excellent results, with all giving a survival rate of 100% 48 h after infection. (Fig. 6b). In addition, mice were infected intraperitoneally with bioluminescent *S. pneumoniae*, two mice were treated with compound **20** (2 mg kg$^{-1}$) and another two mice were untreated as a negative control. Compared with negative control, the treatment group showed dramatically attenuated fluorescence (Fig. 6c). Neutropenic CD-1 mice (via cyclophosphamide), three per group, were infected with $1.5 \times 10^5$ c.f.u. of MRSA ATCC 33591 in their right thigh, and 2 h post infection dosed with compound **20** i.v. at 5, 2.5, and 1.25 mg kg$^{-1}$, or vancomycin 50 mg kg$^{-1}$. Mice were euthanized 24 h later, and the bacterial burden in the thigh determined. Compound **20** had excellent activity, with 5 mg kg$^{-1}$ reducing the bacterial burden 5 log from untreated controls (Fig. 6d).

**Fig. 5** MIC ($\mu g\,ml^{-1}$) for VRE TH4938. **a** Teixobactin was provided by Novobiotic Pharmaceuticals. **b** Teixobactin was synthesized via 5 + 6 strategy. This assay was conducted three times ($n = 3$). Source data are provided as a Source Data file

**Table 2 MIC ($\mu g\,ml^{-1}$) for pathogenic microorganisms[a]**

| Strain | Teixobactin | 20 | 21 | 22 | 24 | 25 | 26 | Ampicillin |
|---|---|---|---|---|---|---|---|---|
| *E. faecalis* ATCC29212 | 1 | 0.25 | 0.5 | 0.25 | 0.25 | – | 0.375 | 1 |
| *E. faecalis* TH4130 | 0.5 | 0.0625 | 0.125 | 0.125 | 0.25 | – | 0.0625 | 16 |
| *E. faecalis* TH4132 | 0.5 | 0.187 | 0.5 | 0.187 | 0.75 | – | 0.187 | 1 |
| *E. faecalis* TH4126 | 0.25 | 0.0625 | 0.187 | 0.125 | 0.25 | – | 0.0625 | >32 |
| VRE TH4939 | 0.25 | 0.125 | 0.187 | 0.25 | 0.5 | – | 0.09 | >32 |
| VRE TH4937 | 0.375 | 0.09 | 0.09 | 0.125 | 0.187 | – | 0.0625 | >32 |
| MRSA BAA-1695 | 0.25 | 0.125 | 0.09 | 0.125 | 0.25 | – | 0.09 | >32 |
| MRSA TH4115 | 0.5 | 0.25 | 0.5 | – | – | 0.25 | 0.25 | 32 |
| *S. aureus* TH4112 | 0.25 | 0.125 | 0.125 | 0.125 | 0.25 | 0.187 | 0.125 | 0.5 |
| *S. pneumoniae* D39 | <0.0625 | <0.0625 | – | – | – | <0.0625 | <0.0625 | 0.125 |
| *S. aureus* ATCC29213[b] | 0.22 ± 0.06 | 0.13 ± 0.07 | – | – | – | – | – | 2 |
| MRSA ATCC33591[b] | 0.11 ± 0.03 | 0.07 ± 0.03 | – | – | – | – | – | >16 |
| *S. agalactiae* ATCC BAA-611 | <0.0625 | <0.0625 | – | – | – | – | – | 0.125 |
| *S. pyogenes* ATCC 19615 | <0.0625 | <0.0625 | – | – | – | – | – | 0.125 |

Source data are provided as a Source Data file
[a]The assay was done three times to confirm results ($n = 3$)
[b]Data represent five independent experiments ± s.d. ($n = 5$)

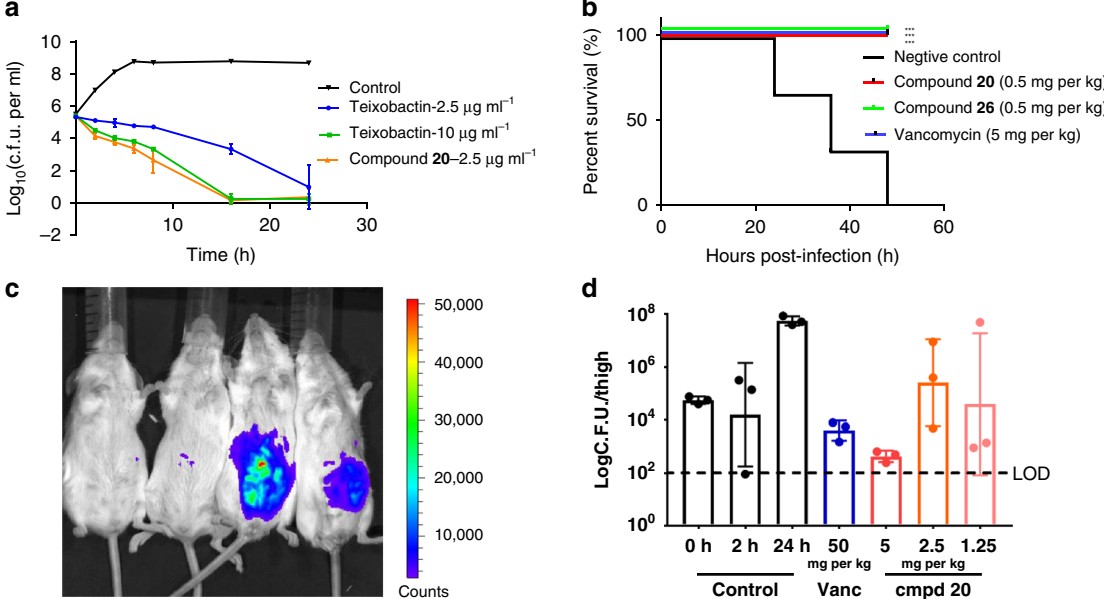

**Fig. 6** Time-dependent killing assay and the in vivo assay of Compounds 20 and 26. **a** Time-dependent killing of VRE ATCC 29212 by teixobactin and Compound **20**. This assay was done twice to confirm the results ($n = 2$). Data represent two independent experiments ± s.d. Source data are provided as a Source Data file. **b** Single dose treatment (i.v., 1 h post infection, six female mice per group) in septicemia protection model using *S. pneumoniae* D39. Survival was depicted 48 h after infection. ***$P < 0.001$ (determined by nonparametric log-rank test). Source data are provided as a Source Data file. **c** BALB/c female mice were infected with bioluminescent *S. pneumoniae* Xen-10 (A66). At left, two mice were treated with compound **20** (2 mg kg$^{-1}$) and the right two mice were negative control. After 48 h, four mice were using IVIS Lumina II. **d** Single dose (i.v., 2 h post infection, three mice per group) treatment with compound **20** and vancomycin in neutropenic mouse thigh infection model using MRSA 33591. For drug-treated animals, thigh colony-forming units (c.f.u.) were determined at 26 h post infection. For controls, c.f.u. in thighs was determined at 0, 2, and 26 h post infection. Source data are provided as a Source Data file. Source data are provided as a Source Data file

**Proposed binding mode of teixobactin.** In 2015, Lewis et al. proposed teixobactin and lipid II could form a complex with the ratio of 2:1[4]. Mu group proposed that the ring motif of two teixobactin molecules bound to the pyrophosphate-MurNAc and the glutamic acid residue of one Lipid II via molecular dynamic study[34]. Very recently, Lewandowski group[35] suggested that residues 2–6 of teixobactin were important for their involvement in aggregation of teixobactin–lipid II complexes with the aid of solid state NMR. Simultaneously, Nowick group described the excellent crystal structure of a derivative of teixobactin, in which sixteen teixobactin derivatives formed an antiparallel β-sheet fibril and two β-sheet fibrils wrap around each other to form a right handed double helix of β-sheets[36]. Due to the lack of support from bioactivity evaluation, they did not ensure the relevance between antiparallel β-sheet of teixobactin and its antibiotic activity. Up to now, the detailed binding mode between teixobactin and lipid II has not been explored. Glide docking has been widely used in structure-based drug development[37]. So we envisioned an integrated approach that combined key amino acids variations of teixobactin and docking to examine the relationship between antiparallel β-sheet and bioactivity of teixobactin, and thus elaborate the detailed mode of action.

First, we set out to explore the correlation between steady antiparallel β-sheet and its bioactivity. For the convenience of description, three consecutive chains of β-sheet of teixobatin were named A, B, and C, respectively (Fig. 7a). The backbone amide of Ile₂ and Ser₃ was important for the formation of β-sheet of AB and BC, respectively. As each of them was involved in the formation of two hydrogen bonds, we changed Ile₂ and Ser₃ to NMe-Ile₂ and NMe-Ser separately to get analogs **27** and **28**, which significantly impeded the intermolecular interaction (Fig. 7b). As expected, CD spectrums of analogs **27** and **28** were

altered and the proportion of antiparallel β-sheet decreased (Fig. 7c, Supplementary Table 3). Moreover, analogs **27** and **28** showed thorough loss of activity (MIC > 25 μg ml$^{-1}$) (Fig. 7d). Despite the overall structural similarity of teixobactin, **27** and **28**, the pattern of these two analogs differed from teixobactin assembly by disturbing intermolecular H-bond interaction of the β-sheet. So we propose that antiparallel β-sheet of teixobactin is essential for its bioactivity.

Next, we embarked on the detailed interaction of teixobactin and lipid II. As teixobactin could target lipid I, lipid II, and C₅₅-PP, according to previous experimental observations and molecular dynamic study[4,34], the pyropyrophosphate group and MurNAc of lipid II were the major binding sites for teixobactin. Given the flexibility of lipid II, truncated lipid II possessing pyrophosphate group and MurNAc moiety was selected as the ligand (Fig. 8a). In the crystal structure[36], one potential pocket was formed by the C-terminal macrocyclic ring of one teixobactin molecule and N-terminus of the other teixobactin molecule. In addition, we confirmed that antiparallel β-sheet of teixobactin was essential for its bioactivity. So antiparallel dimer of compound **26** was regarded as the integral receptor for the following docking (Fig. 8b). Based on the docking, we raise the detailed binding model as the following (Fig. 8c): first, we observed that truncated lipid II bound into the cavity formed by the N-terminus of compound **26** and C-terminal cyclodepsipeptide of the other one. The NH group of backbone of cyclodepsipeptide oriented to the same surface to bind the pyrophosphate group of lipid II. When lactone oxygen atom was replaced with an amide NH group, this NH could form extra hydrogen bond with pyrophosphate group of lipid II. Secondly, there were obvious H-bond and salt bridge between pyrophosphate group and free amine group of N-terminus of the other

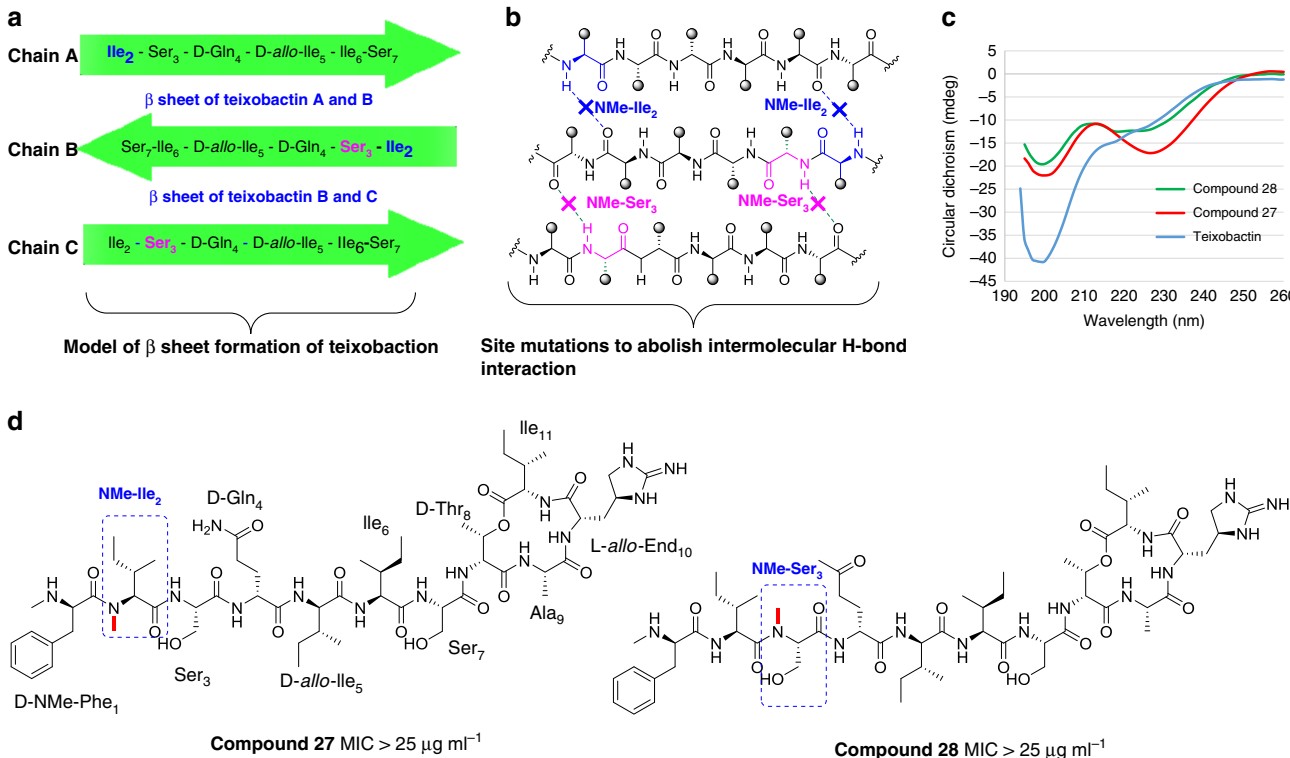

**Fig. 7** Exploring the relationship between antiparallel β-sheet of teixobactin and its bioactivity. **a** Model of b sheet formation of teixobactin and site mutations; **b** site variations to abolish intermolecular H-bond interaction. **c** CD spectrum of teixobactin and analogs **27** and **28**; **d** MIC for MRSA (BAA 1695)

chain. This is consistent with N,N-dimethyl substitution[12] or acetylation[14] of N-terminal amine leading to total loss of activity. Moreover, as the hydrophobic side chains of *N*-Me-D-Phe₁, Ile₂, Ile₅, and Ile₆ lay on the same surface, introduction of biphenyl group at D-Phe position could confer a higher degree of teixobactin assembly formation and anchoring on the membrane. In addition, the shaped guanidine group of L-*allo*-End also bound MurNAc. In fact, as there were two similar cavities in the structure of dimer of teixobactin, truncated lipid II could also bind into the other cavity formed by teixobactin dimer. So far, based on the modifications, variations and docking, we draw the structure–activity relationship of teixobactin as the following (Fig. 8d): First, the cyclopeptide (residues 8–11) and N-terminal amine group were essential for binding lipid II; second, the backbone of Ile₂ and Ser₃ played pivotal role in antiparallel β-sheet formation; third, the introduction of phenyl group on phenylalanine was in favor of assembly of teixobactin on the membrane.

In conclusion, we achieved a one-pot reaction to construct the key building block L-*allo*-End in 30-gram scale and it paved the way for us to develop a convergent strategy (3 + 2 + 6) to achieve gram-scale total synthesis of teixobactin which could promote preclinical applications. Afterwards, in comparison with teixobactin, compounds **20** and **26** were identified as the most potent analogs with 3–8-fold and 2–4-fold greater potency against VRE and MRSA, respectively. They also showed efficacy in a mouse model of *S. pneumoniae* septicemia, and compound **20** was effective against MRSA in a neutropenic mouse thigh infection. We made key amino acid variations to teixobactin, and based on activity and molecular docking we propose that antiparallel β-sheet of teixobactin is important for its bioactivity and an antiparallel dimer of teixobactin is the minimal binding unit for lipid II. Together these studies enhance our understanding of teixobactin and are valuable for further development of lead candidates.

## Methods

**General**. All commercial materials (NJPeptide, Aladdin, J&K Chemical Ltd) were used without further purification. All solvents were analytical grade. CD spectrum and data were collected with Circular Dichroism Spectrometer, model Chirascan Plus. The ¹H NMR ¹³C NMR spectra were recorded on a Bruker 400 MHz spectrometer. High-resolution mass spectra were recorded on Q Exactive. Low-resolution mass spectral analysis was performed with a Waters AQUITY UPLCTM/MS. Analytical HPLC was performed on a SHIMADZU system, using a Vydac 218TP C18 column (5 μm and 4.6 × 250 mm). Semi-preparative HPLC was performed on a SHIMADZU system, using a Vydac 218TP C18 column (10 μm and 10 × 250 mm). Buffer A: 0.1% TFA in acetonitrile; buffer B: 0.1% TFA in H₂O. Mice were from Beijing Vital River Laboratory Animal Technology Co., Ltd. and Charles River Labs. General synthesis procedure and characterization of intermediates of teixobactin and analogs were described in supplementary information.

**Minimum inhibitory concentration assay**. MIC[18] was tested by broth microdilution according to CLSI guidelines. The state of all teixobactin analogs is amorphous, and the compounds are in the salt form of trifluoroacetates. Analogs were stored in DMSO in the concentration of 10 mg/ml. The test medium for most species of bacteria was cation adjusted Mueller–Hinton Broth containing 0.01% polysorbate 80 to prevent analogs binding to plastic surfaces. Analogs were diluted with MHB in a series of concentration for MIC: 1, 0.75, 0.5, 0.375, 0.25, 0.187, 0.125, 0.09, and 0.0625. For the test of Streptococci, MHB contains 3% lysed horse blood (Cleveland Scientific, Bath, OH). The concentration of cell was ~5 × 10⁵ cells ml⁻¹. After incubating at 37 °C in 16–20 h, the MIC was defined as the lowest concentration of antibiotic with no visible growth. Besides the standard strains from ATCC, all the clinical isolates are from 302 hospital. The assay was done three times to confirm results (n = 3).

**MBC assay**. MBC was determined by plating out the dilution representing the MIC and concentrations up to 4× MIC on Mueller–Hinton agar (MHA) plates kept at 37 °C for 24 h. The lowest concentration in which no visible colonies could be detected was taken as the MBC. The assay was done two times to confirm results (n = 2).

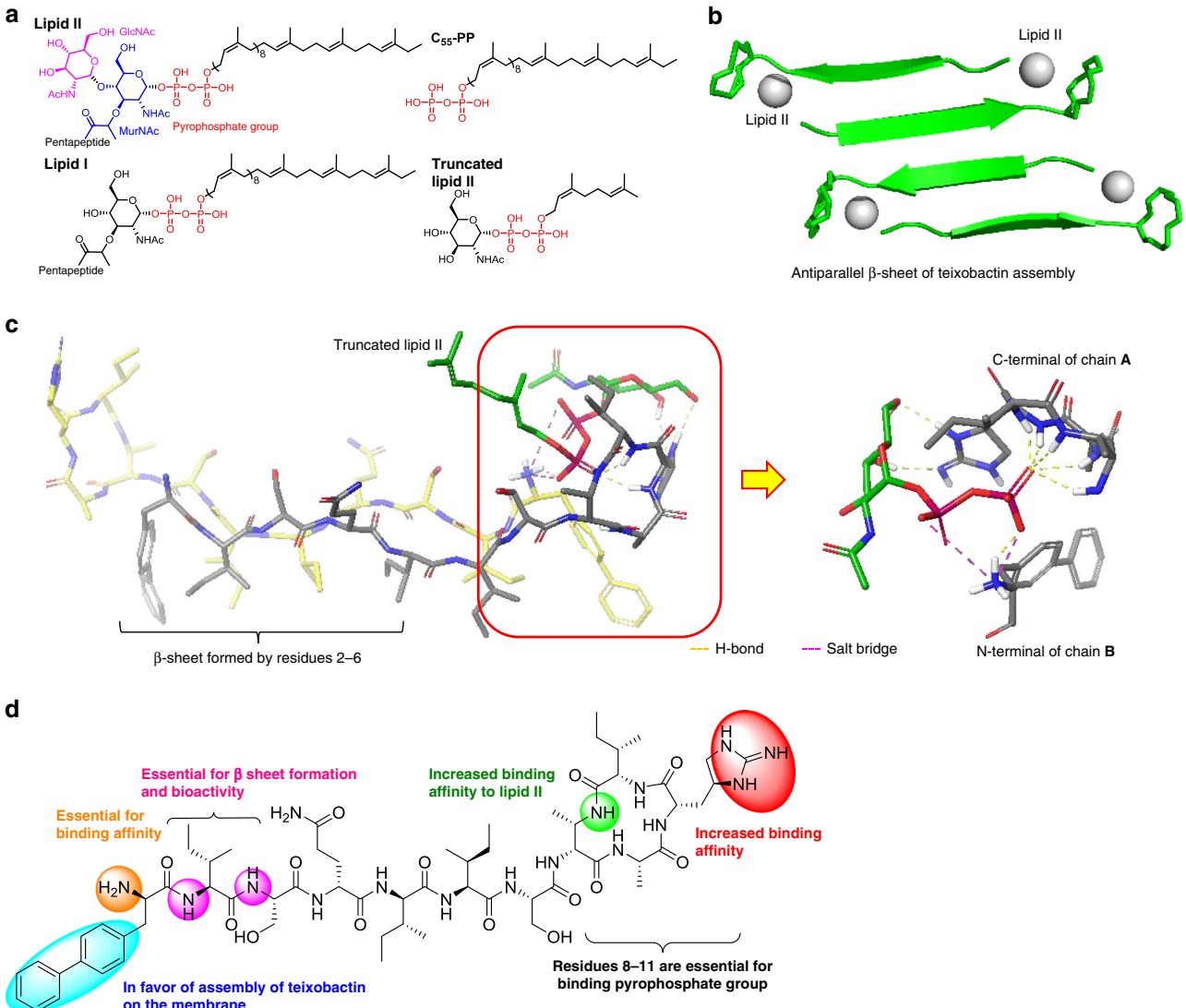

**Fig. 8** Docking structure of compound 26 and truncated lipid II. **a** Structure of lipid I, lipid II, C55-PP, and truncated lipid II; **b** binding model of teixobactin assembly and lipid II; **c** detailed binding mode between dimer of analog 26 and truncated lipid II; step 1: protein preparation (dimer of compound 26); step 2: ligand preparation (truncated lipid II); step 3: receptor grid generation; step 4: ligand docking with Schrödinger Suites Release 2017-1. **d** Summary of SAR of teixobactin analog 26

**Time-dependent killing assay**. Time-kill assay was carried out on VRE ATCC 29212. An overnight culture of cells was diluted 1:1000 in MHB and adjusted to a final inoculum of $10^5$–$10^6$ c.f.u. ml$^{-1}$ in MHB. Bacteria were then challenged with antibiotics at different concentration. teixobactin (2.5 and 10 µg ml$^{-1}$) compound **16** (2.5 µg ml$^{-1}$). At various time points (0, 2, 4, and 8 h), 20 µl aliquots were removed and serially diluted (10–$10^5$-fold dilutions) with sterile phosphate-buffered saline (PBS) and plated onto a MHA plates and incubated for 24 h at 37 °C. Colonies were counted and c.f.u. ml$^{-1}$ was calculated. For analysis of colonies at late stage (16 and 24 h), 1 ml aliquots were removed, centrifuged and resuspended in 100 µl of sterile PBS, and the suspensions was plated on MHA plates, incubated at 37 °C overnight. Colonies were counted and c.f.u. per ml was calculated. This assay was done twice to confirm the results ($n = 2$). Data represents two independent experiments ± s.d.

**Cytotoxicity assay**. The CellTiter 96 AQueous One Solution Cell Proliferation Assay (Promega) was used to determine the cytotoxicity of Compounds **20** and **26**. Exponentially growing HepG2 cells (The HepG2 cell line was kindly provided by Prof. Mengying Lu from 302 hospital in China) were seeded into a 96-well flat bottom plate (2000 cells per well), and incubated at 37 °C. After 24 h, the medium was replaced with fresh medium containing test compounds (5 µL of a twofold serial dilution in DMSO to 495 µL of media). After 48 h of incubation at 37 °C, reporter solution was added to the cells and after 2 h, the A490 nm (OD490) was

measured using Synergy H1 BioTek. Data represent three independent experiments ± s.d. ($n = 3$).

**In Vitro assay**. In vivo assay[4,18]: All animal experiments were done under the guidelines of Laboratory Animal Research Center, Tsinghua University and used an approved animal protocol (16-RY2, PI, Yu Rao) and Northeastern IACUC approved our animal study. (1) Compounds **20**, **26** were tested against *S. pneumoniae* D39 in a mouse septicemia protection assay. The physical state of compounds **20** and **26** were amorphous. They were used in the form of trifluoroacetates. Compounds **20**, **26** and vancomycin were dissolved in a mixture solvent (PBS:Cremophor-EL:DMSO/90:5:5). Species of female mice was C57BL/6J, and they are about 6–8 week old. Each of them was infected with 0.2 mL of bacterial suspension containing about $1 \times 10^4$ c.f.u. (six mice per group) via intraperitoneal injection, and the concentration could result in at least 90% mortality in 48 h after infection. One hour after infection, two experimental groups were treated with compounds **20** and **26** at single intravenous dose of 0.5 mg kg$^{-1}$, respectively. For the two control groups, they were treated with vancomycin or vehicle, respectively. The survival rates were recorded in 48 h. The probability was determined by nonparametric log-rank test. It was analyzed by GraphPad Prism 5.01. (2) For the imaging assay, the species of mice was Balb/c. They were infected with 0.2 mL of bioluminescent *S. pneumoniae* Xen-10 (A66) ($2 \times 10^4$ c.f.u. per mouse, two mice per group) via intraperitoneal injection, 1 h after infection, experimental group was treated with compound **20** at single intravenous doses of

2 mg kg$^{-1}$. Infection control female mice (two per group) were dosed with vehicle. After 48 h, four mice were imaged using IVIS Lumina II. Xenogen Bioware strain *S. pneumoniae* Xen-10 (A66) was bought from Caliper LifeSciences Working Innovation. It harbors a luciferase reporter gene. (3) Female CD-1 mice (6 weeks old, weighing 20–25 g, experimentally naive) from Charles River Labs were made neutropenic via IP injection of cyclophosphamide 4 days (150 mg/kg) and 1 day (100 mg/kg) prior to infection. They were housed three per cage with access to food and water ad libidum. All procedures were performed to IACUC policies and guidelines. *S. aureus* strain ATCC 33591 was prepared by isolation streaking of a glycerol stock onto MHII agar and placing it into a 37 °C incubator overnight. An individual colony was picked from the plate and inoculated into a tube of MHII-broth overnight in a 37 °C shaking incubator. The overnight culture was diluted (OD$_{600}$ = 2.0) 1:1000. Mice were infected with 100 µl of the prepared inoculum into the right thigh with the actual inoculum being $1 \times 10^5$ CFU per thigh (determined from plate counts). Treatment was initiated 2 h post infection. Groups of three mice were treated with the Compound **20** or Vancomycin. All compounds were prepared in a 5% dextrose solution and were delivered via IV injection. Untreated mice were euthanized at 0, 2, and 24 h as controls for bacterial burden. For all control groups and treatment groups, the right thighs were aseptically removed, homogenized in ice cold saline, serially diluted and plated on MHII agar plates. Plates were incubated at 37 °C overnight and bacterial burden was enumerated the following day. Northeastern IACUC approved our animal study.

**CD spectrum collection.** Teixobactin and compounds **27** and **28** were dissolved in mixed solvent (H$_2$O: MeCN: PBS/60:40:1). The final concentration was 0.5 mg/ml. CD spectrum and data were collected with Circular Dichroism Spectrometer, model Chirascan Plus.

**Glide docking.** Docking was performed by Schrodinger (2017-1 released). Step 1: protein preparation: corresponding PDB file 6E00 was downloaded from PDB Bank and loaded into a Maestro interface followed by processed with Protein Preparation Wizard to assign the bond orders and appropriate ionization states and checked for steric clashes. Default parameters were used directly. Only chain A and B, the teixobactin dimer was kept, water, ions, and other chains which were not related to the binding were all removed from workspace. Finally, refinement and minimization were finished by default parameters in Refine Tab using OPLS3 force field. Step 2: ligand preparation: the 3D structure of truncated lipid II was built in in Maestro and then prepared by the LigPrep, neutralized at pH 7.4 by Epik and minimized by OPLS3 force field. Step 3: receptor grid generation: in the constaints of H-bond/Metal, we picked at least one of the amine of the backbone of the lactam (Thr$_8$, Ala$_9$, End$_{10}$, and Ile$_{11}$) could participate in hydrogen bond interaction with truncated lipid II. Step 4: ligand docking: the precision was set to extra precision between the refinement teixobactin dimer and truncated lipid II. The number of poses per ligand to include is set to 5.

**Reporting summary.** Further information on research design is available in the Nature Research Reporting Summary linked to this article.

## Data availability
The authors declare that the data supporting the findings of this study are available within the article (and its Supplementary Information files). And all data is available from the authors upon reasonable request. The raw data underlying Figs. 5, 6a, 6b, 6d, Table 2, Supplementary Fig. 2, Supplementary Tables 1 and 2 are provided as a Source Data file.

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

## Acknowledgements

We acknowledge Novobiotic Pharmaceuticals as the provider of teixobactin; we thank Dr. Xueling Ao from Applied Photophysics Ltd. (Surry, UK) for analyzing CD data; we thank Prof. Yi Xue and Ying Pan from Tsinghua Univerisity for the suggestions about the docking; we thank Qiuye Zhao, Xiuyun Sun for cytotoxicity evaluation of analogs; we thank Yiqing Yang, Xiuyun Sun, and Shi Chen for suggestions in writing; we thank Samantha Niles for assistance with the neutropenic thigh model. This work was supported by National Natural Science Foundation of China (Nos. 81573277, 81622042, 81773567, and 81811530340), National Major Scientific and Technological Special Project for "Significant New Drugs Development" (Nos. 2018ZX09711001 and 2018ZX09301026), and Tsinghua University Initiative Scientific Research Program.

## Author contributions

Y.R. and Y.Z. designed the project; Y.R. and Y.Z. analyzed all the data; Y.Z., F.F. and Z.H.N. synthesized teixobactin and analogs; Y.Z. and F.F. completed the in vitro biological evaluation; Y.Z., H.Y.G. and F.F. performed the in vivo experiments. K.J.M. performed the neutropenic thigh model with MRSA ATCC 33591. K.J.M. performed the MIC and MBC of MRSA ATCC 33591. Y.Z. and L.G.W. conducted the docking. Y.R., J.R.Z. and K.L. designed the bioluminescent *S. pneumoniae* model and neutropenic thigh model with MRSA. The paper was written through contributions of all authors. All authors have given approval to the final version of the paper.

## Additional information

**Competing interests:** The authors declare no competing interests.

