## [Peer Review File · Nature Communications]

Reviewers' comments:

Reviewer #1 (Remarks to the Author):

Review for:

Gram-scale Total Synthesis of Teixobactin, Binding Mode Study and Discovery of More Potent Antibiotics

Yu Zong, Fang Fang, Ligu Wang, Zhihao Ni, Hongying Gao, Jingren Zhang, Yu Rao

Summary:

Rao and coworkers report a convergent strategy for the scalable synthesis of the potent antibiotics teixobactin, enabled by an efficient, transition-metal free intramolecular diamination reaction. Using this strategy, they synthesized several analogs that displayed increased potency and shed light on mechanism of action and structure activity relationships. Two potent analogs were tested in a murine model of infection by *S. pneumoniae*. This work contains important contributions to the understanding of the new antibiotic teixobactin.

Major Critiques:

Animal data often adds to the quality of manuscripts outlining new antibiotics. In this manuscript, however, the animal study only serves to demonstrate that the compounds were safe and effective in the short term for in vivo treatment for an infection that is already susceptible to almost all antibiotics (*S. pneumoniae*). Since teixobactin shows potential to overcome resistant infections, including VRE and MRSA; a model that showed efficacy in a model where vancomycin fails (e.g., VRE) or where β -lactams fail (e.g., MRSA) would have been more impactful. Inclusion of the mouse data as it is doesn't significantly bolster the impact of the manuscript. Additionally, teixobactin analogs have already been tested in mouse models of infection, reducing the novelty of the data in this manuscript.

One of the most interesting portions of this manuscript is the section on mechanism determination. The authors use a truncated version of lipid II in modeling studies to show the hydrogen bonding network that forms upon binding to teixobactin dimers. However, there are no details about any of the modeling protocols or level of theory. This needs to be included in detail in the supporting information, and should also be included in the main text and figure caption.

More importantly, there have already been modeling studies of teixobactin's binding mode to Lipid II, some of which the authors do not reference. See: Liu et al. *Scientific Reports* 2017, 7, 17197.

The authors outline the challenges associated with the synthesis of teixobactin that they aim to overcome (Fig 1). They may have overstated some of these. The L-allo-End does indeed help with activity, but it is certainly not required, and replacement with an arginine leads to analogs that are as active as the parent structure in most strains (see reference 17). Stating it makes a "great contribution" is an overstatement. Additionally, the authors claim that previous syntheses were limited in scale; if they are to do this, they should state what scale the previous syntheses were actually conducted on, to provide a direct comparison. Several previous groups have published syntheses that allowed them to make dozens of analogs.

Several of the structural modifications that the authors have included in their analogs have already been included in several previous teixobactin papers. This includes the biphenyl substitution at the N terminus, the lactone→lactam modification in the cyclic tetrapeptide, and several of the amino acid modifications. The authors should explicitly mention where these modifications has been made before when they mention them in the text.

Figure 4: Compounds 14 and 14-1 have the same structure. I believe 14-1 should be missing a Cbz group.

Minor critiques:

Throughout the manuscript the authors use "mutate" to refer to variations in the amino acids in teixobactin. I think "vary" would be a better fit, since mutation usually implies a genetic and biological origin. Mutate should be avoided when the route is synthetic. See lines 44, 145.

Line 11: "reported" should be "report"

L12: "accomplish a/the gram scale synthesis..."

L13: "2–4-fold increased potency..."
L14: Remove the word "Ultimately"
L22: "eminence" → "emergence"
L26: "from the Payne..."
L27: "lately" → "recently"
L33: "remaining" → "retaining"
L34: "relationships"
L37: "remaining" → "retaining"
L37,38: "the antiparallel..."
L48: delete the word "onto"
L53: delete "as the following."
L68: change "mild" to "moderate"
L79: delete the word "which"
L90: "and we nearly did not observe significant appearance..."
L97: "reductant" → "reducing"
L111: "on the membrane by teixobactin"
L112: "N-terminal" → "N-terminus"
L132: "that lead to 90% mortality" or "that lead to 10% survival"

Supporting information: Page 7: in the synthesis of compound 16, the authors state that the reaction was run under "Ar2" when certainly they mean "Ar".

Conclusion:

Taken as a whole, the major contributions of this paper are: 1) a robust, scalable synthesis of teixobactin; 2) new analogs with increased activity; 3) a model of binding based on strategically selected analogs designed to disrupt binding. These are, without a doubt, significant contributions to the study of an important emerging class of antibiotics. However, much of the manuscript overlaps with previous work, and not all of this work is referenced or described adequately (see Major Critiques above). While this manuscript successfully compiles all of these results into a compelling story, the actual novelty and impact is not as great as it seems as it is currently written.

Some things that could bolster the quality and impact of the manuscript are 1) a mouse model of infection caused by a multidrug-resistant Gram-positive pathogen, especially if an analog shows more efficacy than teixobactin; 2) more details on the molecular modeling, specifically on how other binding modes were ruled out and how this model is actually "new" compared to previous binding and dynamics studies; 3) an expanded MIC table (with more species) and time-kill experiments to determine bactericidal activity.

In its present state, this reviewer believes that the manuscript may be suitable for publication in Nature Communications, but improvements to the manuscript could make it more impactful and thus more appropriate for inclusion in this prestigious journal.

Reviewer #2 (Remarks to the Author):

Zong et al describe the improved synthesis and gram scale yield of teixobactin, a novel natural product antibiotic under pre-clinical development. They also describe a series of new analogs with slight to highly improved in vitro whole cell activity against several gram positive pathogens. Two such analogs showed efficacy in a standard mouse efficacy model when administered by iv injection. The authors go on to test the hypothesis that formation of antiparallel beta sheets are critical for bioactivity. Finally, the authors model binding of teixobactin to lipid 2. Based on their SAR trends and Schrodinger docking models. The strength of the work resides in their detailed and elegant chemistry, which will allow improved synthesis and yield of teixobactin and newer analogs. This will be of interest to those working in natural product and semisynthetic natural product type medicinal chemistry. Overall, these discoveries represent a major step forward and appear novel.

Technically the biological claims are supported by the science. However, there are a few issues that need to be addressed.

1) For the in vitro microbiology, the authors cite a specific fold shift in potency in the text, when in fact a range of values are seen for a given species of pathogen. When referring to overall improvements, the authors should cite the range of values seen or specifically describe a single organism for which they are making the claim. While the potency trends are still visible, it helps to better put the data into context for which analogs are showing more significant shifts in in vitro activity.

2) The authors mostly employed recent clinical isolates for their testing from a local hospital. Although this is an important aspect of testing of new antimicrobial agents, it would be better to use a CLSI reference strain as their "benchmark" (i.e., *E. faecalis* ATCC 29212 – which is included in their data set) as this will allow direct cross comparison in other labs without access to these other recent clinical isolates. ATCC 29212 should probably be listed first in the table.

3) The authors do not mention and QC reference antibiotics (outside of teixobactin) to allow confirmation that the MICs were indeed valid (feel with in the QC range for compounds other than teixobactin or analogs tested herein).

4) Inclusion of high levels (i.e., 0.01%) of polysorbate (tween 80) in the MICs is not standard practice. Were MICs also conducted without polysorbate 80? Can effects of polysorbate on cell potentiation be excluded?

5) Finally, no discussion is given as to why a given compound analog might show anywhere from a marginal two-fold shift in potency (within the error of most MIC assays), to a much more biologically relevant 8 fold shift against any one strain under test? For example, their best compound 26, is 8-fold more potent than teixobactin vs TH4130 and 2.6 fold more potent against ATCC29212, whereas this same compound show 2-fold shifts vs TH4115/4112 but 3-fold shifts vs BAA-1695 Is this due to biochemical differences (i.e., different binding pockets or target related issues)? Or is it likely biological in nature (i.e., reduced interaction with the target)?

6) The authors should list which figure or table they are referring to when describing biological data in the text.

The manuscript would benefit from editorial review by a native English speaker, which would improve the overall readability of the manuscript.

Reviewer #3 (Remarks to the Author):

The authors performed the concise synthesis of allo-End and utilized it in the synthesis of teixobactin analogues in place of D-MePhe with substituted Phe derivatives and teixobactin lactam analogues. They found a biphenylalanine analogue 20 and its lactam analogue 25 are most potent in in vitro assay against VRE and MRSA, better than teixobactin. It is of interest that the lactam analogue is as potent as the parent lactone compound. They were 100% survived in 48 h after i.v. injection to mice and the compound 20 (2 mg/kg) exhibited in vivo efficacy, too.

They also elucidated Ile2 and Ser3 are important to have antiparallel beta-sheet formation that is essential for potent activity because both MeIle2 and MeSer3 derivatives decreased the beta-sheet formation and lost the activity.

Remarkable progress in the research of teixobactin has been made and it could draw attention of the readers. Publication is recommended after minor revision.

1) page 3, Table 1: As NOE was not measured in the intermediate 5 itself, the double-headed arrow in 5 should be removed.

2) page 3, line 63: It might be better to change "guanidinylation" to "guanidinylation of 4".

- 3) page 4, line 76: The authors wrote "the proposed structure of intermediate 5a-2 was indirectly confirmed by NOE between alpha and gamma positions". It would be NOE observation between alpha and gamma protons "in 7".
- 4) page 4, Fig. 3: The double-headed arrow that shows NOE observation should be drawn in compound 7 not in 5a-2. In the Fig. 3b, compound 6a was observed as a MeOH adduct. It seems to be formed during the LC-MS analysis using MeOH as a solvent. It might be better to note the LC-MS analysis was carried out using MeOH as a solvent in the bottom note.
- 5) page 4, line 89: It might be better to change "mild yield" to "moderate yield".
- 6) page 5, Fig 4: The structure of 14-1 has two Cbz groups. I think one or two Cbz group(s) would be removed.
- 7) page 6, line 113: "Compound 20 was 5-fold better than teixobactin." Is 5-fold correct?
- 8) Supporting Information: Please add compound numbers in both HPLC and NMR spectra.

The following information is details about our answers to all reviewer's questions.

Point-by-point replies to the comments by Reviewer 1

Reviewer 1:

1) Animal data often adds to the quality of manuscripts outlining new antibiotics. In this manuscript, however, the animal study only serves to demonstrate that the compounds were safe and effective in the short term for in vivo treatment for an infection that is already susceptible to almost all antibiotics (*S. pneumoniae*). Since teixobactin shows potential to overcome resistant infections, including VRE and MRSA; a model that showed efficacy in a model where vancomycin fails (e.g., VRE) or where b-lactams fail (e.g., MRSA) would have been more impactful. Inclusion of the mouse data as it is doesn't significantly bolster the impact of the manuscript. Additionally, teixobactin analogs have already been tested in mouse models of infection, reducing the novelty of the data in this manuscript.

Reply: Thanks for the reviewer's kind comments. As the reviewer suggested, we have evaluated our best compound **20** in mouse model of infection caused by MRSA ATCC 33591. After treatment with vancomycin and compound **20**, we detected the log C.F.U./thigh. It shows that compound **20** group (5mg/kg) was better than vancomycin group (50 mg/kg) (Please see figure 6d).

2) One of the most interesting portions of this manuscript is the section on mechanism determination. The authors use a truncated version of lipid II in modeling studies to show the hydrogen bonding network that forms upon binding to teixobactin dimers. However, there are no details about any of the modeling protocols or level of theory. This needs to be included in detail in the supporting information, and should also be included in the main text and figure caption. More importantly, there have already been modeling studies of teixobactin's binding mode to Lipid II, some of which the authors do not reference. See: Liu et al. Scientific Reports 2017, 7, 17197.

Reply: Thanks for the reviewer's kind comments. As the reviewer suggested, we cite this important paper in our newly prepared manuscript in in vitro assay part and bind mode part. (please see the reference 34). In the paper published in Scientific Reports, they proposed that two teixobactin bind pyrophosphate-MurNAC and glutamic acid residue of one Lipid II respectively. Undeniably, this is an important discovery. Its structure model also explains experimental observations that the pyrophosphate-MurNAC moiety is the minimal motif for binding. We selected the similar truncated lipid II in our docking. However, this model cannot well explain why the N-terminal amide group is important and methylation in Ile₂ and Ser₃ could result in loss of potency in our study. So we proposed that antiparallel β sheet of teixobactin was essential for its bioactivity.

Unlike other previous binding model, we conducted the ligand docking with antiparallel dimer of teixobactin. We proposed that the antiparallel dimer is the minimal binding motif for lipid II by variations of key amino acids and glide docking. However, we think the simple teixobactin dimer is not stable in solution and the assemble of antiparallel teixobactin could stabilize this binding pocket.

At last, we added the details about modeling protocols into the supporting information.

3) The authors outline the challenges associated with the synthesis of teixobactin that they aim to overcome (Fig 1). They may have overstated some of these. The L-*allo*-End does indeed help with activity, but it is certainly not required, and replacement with an arginine leads to analogs that are as active as the parent structure in most strains (see reference 17). Stating it makes a "great contribution" is an overstatement.

Reply: Thanks for the reviewer's kind comments. As the reviewer suggested, we changed the "great contribution" to "seems to be important". In fact, compared with teixobactin, Arg₁₀-teixobactin actually leads to more than 4-fold loss of potency (Please see J Med Chem, 2018, 61, 2009. Against MRSA, the value of MIC of teixobactin and Arg₁₀-teixobactin is 0.25 and 2 μ g/ml respectively). Actually, one of the best teixobactin analogue Leu₁₀-teixobactin showed high potency, but the solubility was worse than teixobactin. Simultaneously, we synthesized the best analogue D-Arg₄-Leu₁₀-teixobactin as control, which was also slightly weaker than teixobactin (please see supporting information Figure S1).

4) Additionally, the authors claim that previous syntheses were limited in scale; if they are to do this, they should state what scale the previous syntheses were actually conducted on, to provide a direct comparison. Several previous groups have published syntheses that allowed them to make dozens of analogs.

Reply: Thanks for the reviewer's kind comments. As the reviewer suggested, we state what scale the previous syntheses were actually conducted on in our revised manuscript. Previously, all other groups just achieved milligram scale total synthesis of teixobactin (Li group 1.8 mg, Payne group 4.99 mg, Chen group 11 mg). In our study, gram scale synthesis of teixobactin were fulfilled. Additionally, although dozens of analogs have been reported, it was a pity that any bioactivity of analogs remaining L-*allo*-End has not been reported.

5) Several of the structural modifications that the authors have included in their analogs have already been included in several previous teixobactin papers. This includes the biphenyl substitution at the N terminus, the lactone!lactam modification in the cyclic tetrapeptide, and several of the amino acid modifications. The authors should explicitly mention where these modifications has been made before when they mention them in the text.

Reply: Thanks for the reviewer's kind comments. As the reviewer suggested, we cite the related papers in our revised manuscript. According to our previous study based on Lys₁₀-teixobactin, we just observed that increased hydrophobicity of D-Phe improved its activity and lactam was slightly better than lactone against MRSA. As it is the first time to explore the bioactivity of analogs remaining L-*allo*-End, we explore the activity with these modifications.

6) Figure 4: Compounds 14 and 14-1 have the same structure. I believe 14-1 should be missing a Cbz group.

Reply: Thanks for the reviewer's kind comments. As the reviewer suggested, One Cbz was removed in our revised manuscript.

7) an expanded MIC table (with more species) and time-kill experiments to determine bactericidal activity.

Reply: Thanks for the reviewer's kind comments. As the reviewer suggested, we conducted the time-kill experiments and showed an expanded MIC table (Please see figure 6a and Table 2) in our revised manuscript. In the time-dependent killing of *E. faecalis* 29212, compound **20** (**2.5 µg/ml**) was better than teixobactin. The efficiency of compound **20** (**2.5 µg/ml**) was as strong as teixobactin (**10 µg/ml**) (Please see Figure 6a).

8) Throughout the manuscript the authors use “mutate” to refer to variations in the amino acids in teixobactin. I think “vary” would be a better fit, since mutation usually implies a genetic and biological origin. Mutate should be avoided when the route is synthetic. See lines 44, 145. Line 11: “reported” should be “report”

L12: “accomplish a/the gram scale synthesis...”

L13: “2–4-fold increased potency...”

L14: Remove the word “Ultimately”

L22: “eminence” ! “emergence”

L26: “from the Payne...”

L27: “lately” ! “recently”

L33: “remaining” ! “retaining”

L34: “relationships”

L37: “remaining” ! “retaining”

L37,38: “the antiparallel...”

L48: delete the word “onto”

L53: delete “as the following.”

L68: change “mild” to “moderate”

L79: delete the word “which”

L90: “and we nearly did not observe significant appearance...”

L97: “reductant” ! “reducing”

L111: “on the membrane by teixobactin”

L112: “N-terminal” ! “N-terminus”

L132: “that lead to 90% mortality” or “that lead to 10% survival”

Supporting information: Page 7: in the synthesis of compound 16, the authors state that the reaction was run under “Ar₂” when certainly they mean “Ar”.

Reply: Thanks for the reviewer’s kind comments. As the reviewer suggested, they were all corrected in our revised manuscript.

Point-by-point replies to the comments by Reviewer 2

Reviewer 2:

1) For the in vitro microbiology, the authors cite a specific fold shift in potency in the text, when in fact a range of values are seen for a given species of pathogen. When referring to overall improvements, the authors should cite the range of values seen or specifically describe a single organism for which they are making the claim. While the potency trends are still visible, it helps to better put the data into context for which analogs are showing more significant shifts in in vitro activity.

Reply: Thanks for the reviewer's kind comments. As the reviewer suggested, when referring to improvements for the in vitro study, we specifically describe a single organism for which we are making this claim. Simultaneously, we put the MIC data into context in our revised manuscript.

2) The authors mostly employed recent clinical isolates for their testing from a local hospital. Although this is an important aspect of testing of new antimicrobial agents, it would be better to use a CLSI reference strain as their "benchmark" (i.e., *E. faecalis* ATCC 29212 – which is included in their data set) as this will allow direct cross comparison in other labs without access to these other recent clinical isolates. ATCC 29212 should probably be listed first in the table.

Reply: Thanks for the reviewer's kind comments. As the reviewer suggested, the strain of ATCC 29212 is listed first in the table 2. Simultaneously, we added more MIC data for MRSA ATCC 33591 etc. (CLSI reference strain) to allow direct cross comparison in other labs. Actually, we would like to share our clinical isolates with other labs to conduct cross comparison. In addition, we added the MBC data of compound **20** against MRSA (ATCC 33591) in our manuscript. Compound **20** was 2 times better than teixobactin.

Strain	teixobactin	Compound 20
MRSA BAA-1695	0.5	0.125
MRSA ATCC 33591	0.28	0.12

3) The authors do not mention and QC reference antibiotics (outside of teixobactin) to allow confirmation that the MICs were indeed valid (feel with in the QC range for compounds other than teixobactin or analogs tested herein).

Reply: Thanks for the reviewer's kind comments. As the reviewer suggested, Ampicillin was selected as another positive control to validate our value of MIC.

4) Inclusion of high levels (i.e., 0.01%) of polysorbate (tween 80) in the MICs is not standard practice. Were MICs also conducted without polysorbate 80? Can effects of polysorbate on cell potentiation be excluded?

Reply: Thanks for the reviewer's kind comments. As the reviewer suggested, we conducted the MIC without polysorbate 80. The value of MIC (without polysorbate 80) was 4 times higher than the data (with 0.01% polysorbate 80). As the physical property of teixobactin is similar to

Oritavancin, 0.01% polysorbate 80 could prevent drug binding to plastic surfaces (Please see Antimicrob. Agents Chemother. 2008, 52, 1597). In addition, we have confirmed that 0.01% polysorbate 80 could not affect bacteria growth.

Bacteria strain	MIC of Teixobactin (without polysorbate 80)	MIC of Teixobactin (0.01% polysorbate 80)
MRSA BAA-1695	1 µg/ml	0.25 µg/ml

Bacteria strain	Compound 26 (without polysorbate 80)	Compound 26 (0.01% polysorbate 80)
MRSA TH4112	1 µg/ml	0.125 µg/ml

5) Finally, no discussion is given as to why a given compound analog might show anywhere from a marginal two-fold shift in potency (within the error of most MIC assays), to a much more biologically relevant 8 fold shift against any one strain under test? For example, their best compound 26, is 8-fold more potent than teixobactin vs TH4130 and 2.6 fold more potent against ATCC29212, whereas this same compound show 2-fold shifts vs TH4115/4112 but 3-fold shifts vs BAA-1695 Is this due to biochemical differences (i.e., different binding pockets or target related issues)? Or is it likely biological in nature (i.e., reduced interaction with the target)?

Reply: Thanks for the reviewer's kind comments. In our *in vitro* assay, we observed that some kinds of *E. faecalis* are very sensitive to teixobactin and analogues, but the others are not. Other group has also observed the similar phenomenon (Please see J. Med. Chem. 2018, 61, 2009). We think it may be due to the complexity of resistance. As teixobactin targets conserved substrates (lipid II and lipid III), we propose it is not due to the different binding pocket.

6) The authors should list which figure or table they are referring to when describing biological data in the text.

Reply: Thanks for the reviewer's kind comments. As the reviewer suggested, we list the related figure and table when describing biological data in the revised manuscript.

7) The manuscript would benefit from editorial review by a native English speaker, which would improve the overall readability of the manuscript.

Reply: Thanks for the reviewer's kind comments. As the reviewer suggested, a native English speaker has improved the overall readability of this manuscript.

Point-by-point replies to the comments by Reviewer 3

Reviewer 3:

1) page 3, Table 1: As NOE was not measured in the intermediate 5 itself, the double-headed arrow in 5 should be removed.

Reply: Thanks for the reviewer's kind comments. As the reviewer suggested, we double confirmed that we actually measured the NOE of compound **5a**. Please see the NOE spectrum of 5a in our supporting information.

2) page 3, line 63: It might be better to change "guanidinylation" to "guanidinylation of 4".

Reply: Thanks for the reviewer's kind comments. As the reviewer suggested, it was corrected accordingly.

3) page 4, line 76: The authors wrote "the proposed structure of intermediate 5a-2 was indirectly confirmed by NOE between alpha and gamma positions". It would be NOE observation between alpha and gamma protons "in 7". 4) page 4, Fig. 3: The double-headed arrow that shows NOE observation should be drawn in compound 7 not in 5a-2

Reply: Thanks for the reviewer's kind comments. As the reviewer suggested, we double confirmed that we actually measured the NOE of **5a** not in compound **7**. please see the the NOE spectrum of 5a in our supporting information.

4) In the Fig. 3b, compound 6a was observed as a MeOH adduct. It seems to be formed during the LC-MS analysis using MeOH as a solvent. It might be better to note the LC-MS analysis was carried out using MeOH as a solvent in the bottom note.

Reply: Thanks for the reviewer's kind comments. As the reviewer suggested, it is due to the use of MeOH in LC-MS analysis. In our revised manuscript, we added the bottom note.

5) page 4, line 89: It might be better to change "mild yield" to "moderate yield".

Reply: Thanks for the reviewer's kind comments. As the reviewer suggested, it was corrected accordingly.

6) page 5, Fig 4: The structure of 14-1 has two Cbz groups. I think one or two Cbz group(s) would be removed.

Reply: Thanks for the reviewer's kind comments. As the reviewer suggested, it was corrected accordingly.

7) page 6, line 113: "Compound 20 was 5-fold better than teixobactin." Is 5-fold correct?

Reply: Thanks for the reviewer's kind comments. As the reviewer suggested, this description is a little ambiguous. It was corrected accordingly.

8) Supporting Information: Please add compound numbers in both HPLC and NMR spectra.

Reply: Thanks for the reviewer's kind comments. As the reviewer suggested, it was corrected accordingly.

REVIEWERS' COMMENTS:

Reviewer #1 (Remarks to the Author):

The authors have done a very thorough job of addressing all of the reviewers comments. I believe this has resulted in a greatly improved manuscript, which should be of broad interest to the Nature Communications audience. I recommend acceptance and publication.

Reviewer #2 (Remarks to the Author):

I thank the authors for submitting a more complete manuscript.

Reviewer #3 (Remarks to the Author):

The authors have nicely revised the manuscript.
Publication is recommended as it stands.

REVIEWERS' COMMENTS:

Reviewer #1 (Remarks to the Author):

The authors have done a very thorough job of addressing all of the reviewers comments. I believe this has resulted in a greatly improved manuscript, which should be of broad interest to the Nature Communications audience. I recommend acceptance and publication.

Reply: Thanks for the reviewer's kind comments.

Reviewer #2 (Remarks to the Author):

I thank the authors for submitting a more complete manuscript.

Reply: Thanks for the reviewer's kind comments.

Reviewer #3 (Remarks to the Author):

The authors have nicely revised the manuscript.

Publication is recommended as it stands.

Reply: Thanks for the reviewer's kind comments.